# Mortality and Predictive Factors for Death Following the Diagnosis of Interstitial Lung Disease in Patients with Rheumatoid Arthritis: A Retrospective, Long-Term Follow-Up Study

**DOI:** 10.3390/jcm14041380

**Published:** 2025-02-19

**Authors:** Shunsuke Mori, Fumikazu Sakai, Mizue Hasegawa, Kazuyoshi Nakamura, Kazuaki Sugahara

**Affiliations:** 1Department of Rheumatology, Clinical Research Center for Rheumatic Diseases, National Hospital Organization (NHO) Kumamoto Saishun Medical Center, Kohshi, Kumamoto 861-1196, Japan; 2Department of Radiology, Kanagawa Cardiovascular and Respiratory Center, Yokohama, Kanagawa 236-0051, Japan; fmksakai@yahoo.co.jp; 3Department of Respiratory Medicine, Tokyo Women’s Medical University Yachiyo Medical Center, Yachiyo, Chiba 276-8524, Japan; hasemizue@yahoo.co.jp; 4Department of Respiratory Medicine, NHO Kumamoto Saishun Medical Center, Kohshi, Kumamoto 861-1196, Japan; nakamura.kazuyoshi.cy@mail.hosp.go.jp (K.N.); sugahara.kazuaki.ve@mail.hosp.go.jp (K.S.)

**Keywords:** interstitial lung disease, rheumatoid arthritis, mortality, predictive factors for death, HRCT pattern

## Abstract

**Objective:** The aim of this study was to determine mortality and predictive factors for death in patients with rheumatoid arthritis (RA) diagnosed with and without interstitial lung disease (ILD). **Methods:** We retrospectively performed a long-term follow-up study of patients diagnosed with RA at our medical center between April 2001 and June 2023. The diagnosis and classification of ILD were made based on pulmonary high-resolution computed tomography (HRCT), taken at RA diagnosis and during follow-up. **Results:** Among 781 patients with RA, 78 were diagnosed with ILD; all cases except one were subclinical. The most common HRCT pattern was definite usual interstitial pneumonia (UIP) followed by nonspecific interstitial pneumonia (NSIP)/UIP, probable UIP, NSIP, and early UIP. During follow-up (mean of 10.0 years), the crude incidence rate of death (95% confidence interval [CI]) was 7.1 (5.2–10.0) and 1.5 (1.0–1.9) per 100 person-years in RA patients with and without ILD. Poor control of RA activity was associated with increased incidence rates of death. The standardized mortality ratio (95% CI) compared with the general population was 1.32 (1.11–1.53) for all RA patients, 2.09 (1.45–2.73) for RA-ILD patients, and 1.16 (0.95–1.38) for non-ILD RA patients. Lung cancer and respiratory failure were the most common causes of death in RA-ILD patients. The Multivariable Fine-Gray regression analysis revealed that ILD (adjusted hazard ratio [HR] 2.97 [95% CI 1.95–4.53]), advanced age (1.08 per additional year [1.05–1.10]), and low body mass index (3.07 [2.10–4.49]) were strong predictive factors for mortality in RA patients. HRCT patterns did not affect the risk of death in RA-ILD patients. **Conclusions:** Regardless of HRCT pattern, RA-ILD contributes to the increased mortality risk in patients with RA.

## 1. Introduction

Rheumatoid arthritis (RA) is a chronic immune-mediated rheumatic disease that primarily involves multiple synovial joints [1,2]; however, systemic inflammation associated with RA can cause extra-articular damage to various tissues and organs such as the lungs, heart, blood vessels, renal system, skin, nervous system, and eyes [3,4,5]. The incidence of extra-articular manifestations has been decreasing since 2000, but the mortality risk continues to be high in RA patients with extra-articular manifestations and comorbidities [6,7]. In particular, respiratory disease, malignancies, and cardiovascular disease lead to increased mortality in RA patients [8,9,10,11]. Although RA-related mortality has improved over time, there remains a persistent mortality gap between RA patients and the general population or non-RA controls [9,11,12,13].

The respiratory system is a common target of extra-articular manifestations. Although there is extensive heterogeneity among prevalence studies of lung involvement in RA, interstitial lung disease (ILD) has the greatest estimated prevalence, followed by airway disease, pleural effusion, and rheumatoid nodules [14,15,16]. Subclinical and clinical ILD are increasingly recognized throughout the entire RA disease course; interstitial lung abnormalities can be detected in up to 60% of RA patients by high-resolution computed tomography (HRCT), and symptomatic ILD likely occurs in 5% to 17% of patients [14,16,17]. Advanced age, male sex, smoking, and RA-related autoantibodies are recognized as patient-level risk factors for developing RA-ILD [18]. Despite advances in the management of RA, ILD may continue to be a major contributor to mortality in RA. Several population-based cohort studies have shown an increased mortality associated with RA-ILD compared with non-ILD RA [19,20,21,22,23,24,25]. Other comparison studies between RA patients and the general population or non-RA patients also suggest that ILD contributes to excess mortality in RA [11,26,27,28,29]. Delayed diagnosis, lack of biological or targeted drugs for RA-ILD, the existence of rapidly progressive fibrosis, and serious complications (such as lung cancer, pulmonary infection, respiratory failure, and pulmonary hypertension) cause poor outcomes associated with RA-ILD [6,15,30].

It is important to explore whether clinical characteristics and HRCT patterns could predict death outcomes for RA-ILD patients. Although a variety of HRCT patterns are seen in RA-ILD patients, the most common patterns may be usual interstitial pneumonia (UIP) and nonspecific interstitial pneumonia (NSIP) [14,16,31]. However, UIP and NSIP patterns on a HRCT scan often overlap, and indeterminate patterns for UIP can be observed [32,33,34,35], which may obscure the impact of HRCT patterns on mortality in RA-ILD patients in previous studies [36,37,38].

To evaluate mortality, cause of death, and predictive factors for death in RA patients with ILD, we retrospectively conducted a long-term follow-up study of patients diagnosed with RA at our rheumatology department between April 2001 and June 2023. The diagnosis and classification of ILD were made based on pulmonary HRCT, taken at RA diagnosis and during follow-up. Mortality estimates and causes of death were compared between RA patients with and without ILD. Using Fine-Gray competing risk regression analysis, we examined the effect of each baseline characteristic on death outcome over time. We also calculated standardized mortality ratios (SMRs) in our RA cohort with and without ILD compared with the general population in Japan. Additionally, we explored the impact of RA activity control on poor outcomes based on the 28-joint disease activity score using C-reactive protein (DAS28-CRP).

## 2. Materials and Methods

### 2.1. Patients

We used a real-world database including patients who received a diagnosis of RA at the rheumatology department of National Hospital Organization (NHO) Kumamoto Saishun Medical Center between April 2001 and June 2023. Participants in this study were required to meet the 2010 American College of Rheumatology (ACR)/European League Against Rheumatism (EULAR) criteria for diagnosis of RA [39]. Participants were also required to be 18 years of age or older. We excluded patients from this study if they had any of the following histories prior to RA diagnosis: (1) other collagen vascular diseases/autoimmune diseases; (2) exposure to asbestos or silica; or (3) thoracic radiation for cancer therapy.

### 2.2. Study Design

The diagnosis of RA-associated ILD (RA-ILD) was made based on pulmonary HRCT taken at RA diagnosis and during follow-up, irrespective of the presence or absence of clinical symptoms. We reviewed patient medical records and collected clinical data when the diagnosis of RA-ILD was made (baseline characteristics of patients), which included demographic characteristics; RA-related data (duration of joint signs and symptoms, DAS28-CRP, Steinbrocker stage, anticyclic citrullinated peptide antibodies [anti-CCP], and rheumatoid factor [RF]); smoking history; body mass index (BMI); and comorbidities such as type 2 diabetes, malignancy history, and cardiovascular disease (CVD) history. CVD included angina pectoris, cardiac infarction, and stroke. The entry year was also recorded. For RA patients without ILD, clinical data at the time of RA diagnosis were used as the baseline characteristics. For patients who were diagnosed with RA or RA-ILD before a commercially available enzyme-linked immunosorbent assay (ELISA) kit was available, we used sera that were collected at the time of each diagnosis and stored at −80 °C. Detailed methods for the measurement of anti-CCP are described in our previous study [40].

Follow-up started on the day of the HRCT-based diagnosis of ILD (for RA-ILD patients) or RA diagnosis (for RA patients without ILD) and ended with death, loss to follow-up, or the last follow-up visit before 31 December 2023, whichever occurred first. Patients who missed two or more scheduled clinical appointments were classified as lost to follow-up. We determined the cause of death according to each treating physician’s judgment.

### 2.3. Pulmonary Function Tests and ILD Severity

For RA-ILD patients, we collected data of pulmonary function tests (PFTs) at the time of RA-ILD diagnosis. The forced vital capacity (FVC), forced expiratory volume in the first second (FEV_1_), maximal mid-expiratory flow (MMF; also called forced expiratory flow from 25% to 75% of vital capacity [FEF_25–75_]), and diffusing capacity of carbon monoxide (DL_CO_) were measured using a rolling-seal type of spirometer. The results were expressed as a ratio of the measured to the predicted values (% predicted). The predicted values were calculated based on the age, sex, and height of the individual. An abnormal PFT was defined as a % predicted value < 80% for each PFT. The FEV_1_/FVC ratio was also calculated; an abnormal FEV_1_/FVC ratio was defined as <70%. Additionally, we collected data on exercise-induced oxygen desaturation, which was defined as a drop in percutaneous oxygen saturation (SpO_2_) < 90% during a 6 min walk, at the time of diagnosis of RA-ILD. Disease severity staging at that time was determined based on the arterial partial pressure of oxygen (PaO_2_) at rest and SpO_2_ during the 6 min walk test according to the Japanese Respiratory Society Guidelines [41,42].

### 2.4. Classification of HRCT Pattern of ILD

Pulmonary HRCT images of ILD patients taken at RA diagnosis and during subsequent follow-ups were collected from our database and viewed in random order and independently by three observers (one board-certified radiologist [F. Sakai] and two board-certified pulmonologists [M. Hasegawa and K. Nakamura]) who were blinded to the patients’ clinical status and PFT results. Final decisions were made by discussing whether there was disagreement among the three experts after the first assessment. We did not determine the inter-reader agreement in assessing the HRCT features by kappa statistics because all observers were very experienced at identifying abnormalities on pulmonary HRCT, especially of parenchymal and airway lung diseases. HRCT abnormalities included the following findings: bronchiectasis or bronchiolectasis, bronchial wall thickening, centrilobular micro-nodules and a branching structure, cysts, ground-glass opacity (GGO), intralobular reticular opacity, airspace consolidation, honeycombing, traction bronchiectasis, architectural distortion, and emphysema. The distribution of HRCT abnormalities was evaluated according to six zones (upper, middle, and lower zones; involvement in at least 5% of any lung zone). We also examined whether these abnormalities had a predominantly subpleural or non-subpleural distribution.

Each patient with RA-ILD was classified as having one of the following four HRCT patterns: definite UIP, probable UIP, indeterminate for UIP, and alternative diagnosis according to the updated official American Thoracic Society (ATS)/European Respiratory Society (ERS)/Japanese Respiratory Society (JRS)/Latin American Thoracic Society (ALAT) clinical practice guideline for idiopathic pulmonary fibrosis (IPF) [43,44]. In this study, ILD patients with coexisting airway abnormalities such as bronchiolitis or bronchiectasis on HRCT scans were classified as RA-ILD, because a high prevalence of airway abnormalities has been observed in the RA population [32,45]. We further classified the “indeterminate for UIP” pattern into early UIP and NSIP/UIP based on the modified HRCT classification for RA-ILD that was reported by Yamakawa et al. [46]. In brief, honeycombing with a subpleural, basal-predominant distribution is a distinguishing feature of the definite UIP pattern. It can be seen with or without peripheral traction bronchiectasis/bronchiolectasis. Subpleural, basal-predominant reticulation with peripheral traction bronchiectasis/bronchiolectasis was considered the probable UIP pattern. The early UIP pattern was assigned when there were HRCT features consisting of subpleural, basal-predominant mild reticulation or GGO without traction bronchiectasis/bronchiolectasis. If central or diffuse distribution of GGO or reticulation as the component of NSIP and subpleural reticulation with traction bronchiectasis/bronchiolectasis, with or without honeycombing as the component of UIP, were both observed, the NSIP/UIP pattern was assigned. HRCT features showing lower lobe-predominant, central or diffuse GGO with limited or no reticulation and the absence of honeycombing were consistent with the NSIP pattern. In the present study, all patients who had an alternative diagnosis were finally diagnosed as NSIP.

### 2.5. DMARD Use and Control of RA Activity During Follow-Up

The exposure to disease-modifying antirheumatic drugs (DMARDs) during follow-up were examined using patient medical records. Based on a mean of DAS28-CRP values measured for 6 months prior to the end of follow-up, each patient was classified into a group with good control of RA activity (remission or low disease activity [DAS28-CRP < 2.7]) and a group with poor control of RA activity (moderate or high disease activity [DAS28-CRP ≥ 2.7]).

### 2.6. Statistical Analysis

The mean and standard deviation (SD) were used as descriptive statistics for data with continuous distribution, which included non-normally distributed data [47,48]. Number (percentage) was used for categorical data. We compared the baseline characteristics between the two patient groups using Fisher’s exact probability test for categorial variables and the independent-measures *t*-test for continuous variables. There were no missing baseline measurements.

Crude incidence rates of death and 95% confidence intervals (CIs) were calculated by dividing the number of death cases by the number of corresponding follow-up person-years (PYs).

SMRs and 95% CIs in RA patients with and without ILD, in male and female patients separately, compared with the general population in Japan, were calculated as the ratio of the observed number of deaths in our cohort divided by the expected number of deaths in the reference population from 2001 to 2023. For the reference population, we used mortality data in the Japanese general population reported by Japanese Health and Wealth (http://www.stat.go.jp/english/index.html (accessed on 2 December 2024)). The expected number of deaths was calculated based on age- and sex-specific population mortality rates at each calendar year for patients at risk during follow-up.

We used the cumulative incidence function (CIF) to estimate the probability of death before a given time, because we considered the presence of competing risks (i.e., loss to follow-up). Gray’s test was used to test the equality of CIF mortality estimates over time between RA patients with ILD and those without ILD [49]. For the same reason, Fine-Gray competing risk regression analysis was used to calculate the adjusted hazard ratios (HRs) with 95% CIs for death outcome over time. As predictor variables, we selected a set of baseline characteristics based on previous knowledge about the clinical relevance and importance of each variable, which included demographic data, RA-related data, RA-ILD diagnosis, smoking history, BMI, comorbidities, and entry year. For RA-ILD patients, HRCT patterns (definite UIP, probable UIP, indeterminate for UIP, and NSIP patterns) and HRCT abnormalities (honeycombing, traction bronchiectasis, and emphysema) were included as predictor variables. Predictor variables with *p*-values < 0.10 in univariable analyses were used in a multivariable Fine-Gray regression analysis. A forced entry procedure was selected in the multivariable analysis.

Two-sided *p*-values < 0.05 were considered to indicate statistical significance. For all statistical calculations, we used PASW Statistics version 27 (SPSS Japan Inc., Tokyo, Japan) and Easy R (Saitama Medical Center, Jichi Medical University, Saitama, Japan) [50].

## 3. Results

### 3.1. Baseline Characteristics of RA Patients with and Without ILD

A total of 781 patients were included in this study (Appendix A). Based on the HRCT findings taken at RA diagnosis and subsequent follow-up, 78 patients were diagnosed with RA-ILD (10.0%) and 703 were diagnosed without ILD. Clinical characteristics at baseline are shown in Table 1. No patients had missing data for baseline characteristics. RA-ILD patients had a mean age of 70.6 years, which was significantly higher than RA patients without ILD (62.4 years) (*p* < 0.001). Male patients were predominant in the RA-ILD group compared with the group without RA-ILD (55.1% vs. 22.6%, *p* < 0.001). The mean duration from the onset of joint symptoms was similar in patients with and without ILD (3.6 vs. 3.7 years). Approximately 90% of patients had high or moderate disease activity at baseline, and no significant difference in DAS28-CRP levels was observed between patient groups. Rates of anti-CCP-positive patients and RF-positive patients were significantly higher in RA-ILD patients than in RA patients without ILD (anti-CCP-positive, 96.2% vs. 86.6%, *p* = 0.011; RF-positive, 97.4% vs. 86.3%, *p* = 0.003). Smoking history was also significantly associated with the presence of RA-ILD (41.0% in RA-ILD patients vs. 10.0% in RA patients without ILD, *p* < 0.001). Additionally, RA-ILD patients more frequently had a history of CVD compared with RA patients without ILD (17.9% vs. 8.3%, *p* = 0.011).

### 3.2. Baseline Features of RA-ILD

Characteristics of patients with ILD are shown in Table 2. Among 78 RA-ILD patients, 65 (83.3%) were diagnosed with ILD at the same time as their RA diagnosis. ILD preceded their RA diagnosis in three patients (3.8%). The most common HRCT pattern was definite UIP (37.2%) followed by NSIP/UIP (26.9%), probable UIP (17.9%), NSIP (11.5%), and early UIP (6.4%). Honeycombing was observed in all patients with definite UIP and in 10 patients with NSIP/UIP. Traction bronchiectasis was seen in all patients with probable UIP and in most patients with definite UIP or NSIP/UIP. Emphysema was observed in 46.2% of RA-ILD patients. Mean values of the % predicted for FVC and FEV_1_ were within the normal range. The mean % predicted for DL_CO_, an index of restrictive interstitial disease, was 83.2%. The FEV_1_/FVC ratio, a sensitive index of overall airway obstruction, was 125.5%. It is worth noting that the mean % predicted for MMF, a specific index of small airway function, was 70.4%, indicating that obstructive changes in small airways are present in RA-ILD patients. Only one patient showed desaturation with exercise (SpO_2_ < 90% during a 6 min walk) and disease severity stage III. This patient had clinical symptoms at the time of ILD diagnosis.

### 3.3. Mortality in RA Patients with and Without ILD

After the diagnosis of RA-ILD or RA was made, patients were followed for a mean of 10.0 years. As shown in Table 3, 128 patients (16.4%) were lost to follow-up. Death events occurred in 152 patients (19.5%; 41 RA-ILD patients [52.6%] and 111 RA patients without ILD [15.8%]). The crude incidence rate of death was significantly higher in RA-ILD patients than in RA patients without ILD (7.1 per 100 PYs [95% CI, 5.2–10.0] for RA-ILD patients and 1.5 per 100 PYs [95% CI, 1.0–1.9] for patients without RA-ILD). There were significant differences in the CIF-estimated cumulative incidence of death over time between patient groups (*p* < 0.001 with Gray’s test). For example, the cumulative incidence (mortality rate) at 5 years was estimated to be 18.6% (95% CI, 10.5–28.6) for RA-ILD patients and 3.4% (95% CI, 2.2–5.0) for RA patients without ILD. The CIF plots for the probability of death events in RA-ILD patients grouped according to HRCT patterns are shown in Appendix A. There was no significant difference in CIF mortality estimates among these four patterns (*p* = 0.57 with Gray’s test with post hoc Holm test).

The SMRs in our RA cohort compared with the general population in Japan were 1.32 (95% CI, 1.11–1.53) for overall RA patients, 2.09 (95% CI, 1.45–2.73) for RA-ILD patients, and 1.16 (95% CI, 0.95–1.38) for RA patients without ILD.

### 3.4. Cause of Death in RA Patients with and Without ILD

Among RA-ILD patients, 13 (31.7%) died due to malignancy, primarily lung cancer (Table 4). Twelve RA-ILD patients (29.3%) developed respiratory failure due to an acute exacerbation of ILD. Six patients (14.6%) died from serious infections. Among RA patients without ILD, malignancy was the most common cause of death (30.6%), followed by cardiovascular events (22.5%) and serious infection (13.5%). Death due to lung cancer and respiratory failure occurred more frequently in RA-ILD patients than in RA patients without ILD (lung cancer, 24.4% vs. 2.7%, *p* < 0.001; respiratory failure, 29.3% vs. 5.4%, *p* < 0.001). The rate of cardiovascular events as the cause of death was lower in RA-ILD patients than in RA patients without ILD (4.9% vs. 22.5%, *p* = 0.015). Overall malignancies or serious infection caused death at similar rates between RA patients with and without ILD.

### 3.5. DMARD Use During Follow-Up and RA Control Within 6 Months Prior to the End of Follow-Up

As shown in Appendix A, approximately 90% of RA patients without ILD received MTX as monotherapy or in combination therapy with other DMARDs during follow-up, whereas more than 40% of RA-ILD patients never received MTX, most likely because of concerns over ILD exacerbation. Instead of MTX treatment, RA-ILD patients were more often prescribed other csDMARDs. Additionally, IL-6 inhibitors and abatacept were prescribed more frequently for RA-ILD patients than for RA patients without ILD. The rates of the TNF inhibitor and JAK inhibitor use were similar between the two patient groups. Exposure duration to MTX was longer in RA patients without ILD than in RA-ILD patients. Exposure duration to other DMARDs did not differ significantly between RA-ILD patients and RA patients without ILD. Limited to RA-ILD patients, MTX users more often received other csDMARDs compared with non-MTX users. The rate of TNF inhibitor use was significantly higher among MTX users because TNF inhibitors are recommended to be used concomitantly with MTX (Appendix A).

The status of RA control in each patient for 6 months before the end of follow-up was determined based on the mean DAS28-CRP value measured during this period and used to explore the influence of RA activity on mortality. The rate of poor control of RA activity was significantly higher in RA patients with ILD than in those without ILD (19.2% vs. 8.8%, *p* = 0.008), which may be explained by the restriction of DMARD use, especially MTX use, in RA-ILD patients. As shown in Table 5, the crude incidence rate of death events was significantly higher in the poor RA control group compared with the good RA control group (8.3 per 100 PYs [95% CI, 6.4–11.0] vs. 1.3 per 100 PYs [95% CI, 1.1–1.6]). These data suggested that the poor control of RA activity was associated with an increased risk of death in RA patients. There was no significant difference in incidence rates of death between patients with and without RA-ILD in the poor RA control group. In contrast, RA-ILD patients had a higher incidence rate of death than RA patients without ILD in the good RA control group. Thus, the increased mortality in RA-ILD patients was not explained simply by poor RA control due to the restriction of DMARD use.

### 3.6. Predictive Factors for Death in Overall RA Patients and RA-ILD Patients

The results from the univariable and multivariable Fine-Gray regression analyses for overall RA patients are shown in Table 6. Through multivariate modeling, age (adjusted HR 1.08 per additional year [95% CI, 1.05–1.10], *p* < 0.001), RA-ILD (adjusted HR 2.97 [95% CI, 1.95–4.53], *p* < 0.001), BMI < 17 (adjusted HR 3.07 [95% CI, 2.10–4.49], *p* < 0.001), anti-CCP-positive (adjusted HR 2.87 [95% CI, 1.25–6.56], *p* = 0.013), and male sex (adjusted HR 1.61 [95% CI, 1.02–2.53], *p* = 0.042) at baseline were identified as independent predictive factors for death.

The results for RA-ILD patients are shown in Table 7. Age (adjusted HR 1.08 per additional year [95% CI, 1.03–1.13], *p* = 0.002), BMI < 17 (adjusted HR 5.29 [95% CI, 2.28–12.27], *p* < 0.001), and CVD (adjusted HR 0.26, 95% CI, 0.08–0.85], *p* = 0.030) at baseline were independent predictive factors for mortality. Neither HRCT patterns nor HRCT abnormalities were identified as factors associated with death outcomes.

## 4. Discussion

In the present study, we showed that approximately 10% of RA patients developed ILD during the course of the disease. The most common HRCT pattern was definitive UIP followed by NSIP/UIP, probable UIP, NSIP, and early UIP. Lung cancer and respiratory failure were the most common causes of death in RA-ILD patients. During follow-up over a mean of 10.0 years, the crude incident rate of death was 5 times higher in RA-ILD patients than in RA patients without ILD, although all ILD cases except one were subclinical at the time of first diagnosis. The SMRs (95% CI) compared with the general population were 1.32 (1.11–1.53) for all RA patients, 2.09 (1.45–2.73) for RA-ILD patients, and 1.16 (0.95–1.38) for RA patients without ILD. Patients whose RA activity was poorly controlled had a significantly higher incidence rate of death compared with patients in the good RA control group. According to multivariable Fine-Gray regression analysis, ILD, advanced age, and low BMI were predictive factors for increased mortality in RA patients. HRCT patterns did not affect the risk of death in RA-ILD patients.

Using data from the French national claims database from 2013 to 2018, Juge et al. showed that RA-ILD patients had a 3.4-fold increase (95% CI, 3.1–3.9) in mortality rate compared with RA patients without ILD, especially in patients aged <75 years and in male patients [25]. Using the Medicare claims database from 2008 to 2017, Sparks et al. reported that RA-ILD had an HR of 1.66 (95% CI, 1.60–1.72) for total mortality compared with RA without ILD [21]. In a population-based cohort study in Denmark between 2004 and 2016, the HR ratio for death during follow-up > 5 to 10 years was 2.7 times higher (95% CI, 1.9–3.9) for RA-ILD compared with RA without ILD and it was higher in male patients [20]. Using data from the US Centers for Disease and Control Prevention, Gao et al. showed that the mortality rate ratio of RA-ILD to RA was 6.90 from 1999 to 2003 and 6.49 from 2014 to 2018, and they suggested that RA-ILD may contribute to the reduced life expectancy [22]. In the present study, we showed that the HRCT-based diagnosis of ILD, albeit subclinical, was a strong predictive factor for increased mortality in RA patients (adjusted HR 2.97 [95% CI, 1.95–4.53]). Thus, RA-ILD is associated with an increased mortality risk compared with RA without ILD.

In a national, matched-cohort study with the United States Veterans Health Administration database from 2000 to 2017 (RA vs. non-RA), Johnson et al. showed that ILD was the cause of death most strongly associated with RA (adjusted HR 3.39 [95% CI, 2.88–3.99]) [11]. In a retrospective cohort study using the Australian Rheumatology Association database from 1995 to 2020, Black et al. showed that ILD is an important cause of death in RA patients (cause-specific SMR 7.64 [95% CI, 3.98–14.69]) [29]. In a cohort study using the Korean National Statistical Office from 2001 to 2007, Kim et al. showed that death from ILD was significantly higher in RA patients (cause-specific SMR 18.18 [95% CI, 2.20–65.64]) [26]. In the present study, SMRs compared with the Japanese general population were 1.32 (95% CI, 1.11–1.53) for all RA patients and 2.09 (95% CI, 1.45–2.73) for RA-ILD patients. There was no significant difference in mortality rates in RA patients without ILD compared with the general population. These findings indicate that ILD contributes to excess death in RA patients compared with the general population or non-RA patients.

The effect of HRCT pattern on mortality in RA-ILD patients remains controversial. Some studies have shown that the UIP pattern is associated with an increased mortality risk in RA-ILD patients [51,52,53,54,55,56,57,58,59]. However, other studies have shown that the extent of lung involvement or severity of lung disease, rather than the baseline UIP pattern, independently predicts mortality [60,61,62,63,64,65]. Another study showed that survival in RA-ILD patients did not differ between the UIP and NSIP patterns [66]. Recently, Yamakawa et al. developed a modified HRCT classification for RA-ILD according to the latest guideline of IPF, in which the “indeterminate for UIP” pattern was further classified into early UIP and NSIP/UIP patterns [46]. Using the modified HRCT classification, Yamakawa et al. showed that their RA-ILD cohort included NSIP/UIP (27%), definite UIP (21%), probable UIP (20%), NSIP (14%), and early UIP (3%). In that study, the HRCT pattern was not a predictive factor for mortality in a multivariable Cox regression analysis [46]. In the present study, we followed the modified HRCT classification by Yamakawa et al. and found that in our RA cohort, the most common HRCT pattern was definite UIP (37.2%), followed by NSIP/UIP (26.9%), probable UIP (17.9%), NSIP (11.5%), and early UIP (6.4%). The HRCT pattern was not related to mortality risk in our RA-ILD patients. In both studies, most patients with the “indeterminate for UIP” pattern were classified as having the NSIP/UIP pattern. In other previous studies, the NSIP/UIP pattern might have been simply diagnosed as NSIP or UIP, which could have resulted in different conclusions regarding the effect of HRCT pattern on mortality risk in RA-ILD patients. In our statistical analyses, the presence of ILD was a strong predictive factor for death, and the NSIP pattern did not have a significantly better prognosis than the UIP, NSIP/UIP, or early UIP pattern, which suggests that, regardless of HRCT pattern, RA patients with ILD are at increased risk of death. Jacob et al. showed that the individual extents of fibrosis, reticulation, and honeycombing on HRCT scans could independently predict the development of progressive fibrosis and poor outcomes in RA-ILD patients [67]. In this respect, HRCT examinations for ILD at RA diagnosis and during follow-up can provide insight into the extents of these abnormalities.

There is some evidence of associations between RA disease activity and progression of ILD/death outcome in RA-ILD patients [68]. In a retrospective, single-center cohort study of RA-ILD patients in China, Chai et al. showed that high RA disease activity and an advanced extent of lung involvement were independent risk factors for the progression of radiologic fibrosis [63]. In a multicenter, prospective cohort study of US veterans with RA-ILD, Brooks et al. indicated that uncontrolled RA disease activity and the functional status of joints, as well as the severity of lung disease, were associated with increased mortality in this patient population [69]. In a retrospective, single-center cohort study of RA-ILD patients in China, Liu et al. showed that in addition to the UIP pattern, high RA disease activity and advanced fibrosis were independent risk factors for RA-ILD progression [58]. In a prospective study with data from the Korean Rheumatoid Arthritis-Interstitial Lung Disease cohort, Chang et al. found no association between the risk of deterioration of ILD/mortality and RA disease activity [64]. In the present study, the crude incidence rate of death events was significantly higher in the poor RA control group compared with the good RA control group. Thus, the poor control of RA activity was associated with the increased risk of death. The restriction of DMARD use in RA-ILD patients may have resulted in the higher rate of poor control of RA activity in this patient group compared with RA patients without ILD. Concerns about lung injury are greatest in patients receiving MTX, but a causal effect between MTX and fibrotic ILD has not been established [30,70]. The overall risk of worsening lung disease attributable to MTX is currently unclear. Recent studies do not support the idea that RA-ILD progression is increased in patients receiving MTX [71,72]. The 2021 ACR guidelines for treatment of RA include a conditional recommendation for MTX for the treatment of RA patients with clinically diagnosed mild and stable airway or parenchymal lung disease who have moderate to high disease activity [73]. The appropriate use of MTX considering RA disease activity would be important for the management of RA-associated ILD. The use of IL-6 inhibitors and abatacept may be considered in the treatment for RA-ILD patients because concomitant MTX use is not necessarily required for these DMARD therapies.

In the present study, all ILD cases, except one, were subclinical at baseline. Nevertheless, the mortality rate was 2.1 times higher in RA-ILD patients compared with the general population during long-term follow-up with a mean of 10 years. Among RA-ILD patients, the most common cause of death was respiratory failure due to acute exacerbation of ILD and lung cancer. Some RA patients with ILD are known to experience acute exacerbation, with sudden progression of fibrosis and short-term mortality. Unfortunately, it is difficult to predict which patients with RA-ILD are at a high risk of ILD progression. The regular monitoring of RA-ILD patients is important for the prompt identification of ILD progression and early intervention to preserve lung function [68,74]. Recently, we reported that the standardized incidence ratios of lung cancer in our RA cohort compared with the general population was 2.53 (95% CI, 1.29–3.77) for male patients. The presence of ILD was a strong predictive factor for lung cancer occurring in RA patients over time [75]. During follow-up, increased lung cancer-related mortality was observed in RA patients compared with non-RA patients (crude incidence rate: 0.29 per PY vs. 0.10 per PY). The presence of combined pulmonary fibrosis and emphysema was identified as an independent predictive factor for lung cancer-related mortality [76]. Non-surgical cancer treatment includes chemotherapy and radiation therapy, which are well known to exacerbate pulmonary fibrosis. Limited options for lung cancer treatment and treatment-associated complications may be associated with the increased lung cancer-related mortality observed in RA-ILD patients.

There are at least four potential limitations concerning the results of this study. First, this was a retrospective long-term observational study, in which there might be some difficulty in tracing subjects. However, the electronic medical records in the database of our medical center allowed us to obtain accurate clinical and HRCT data obtained at RA-ILD diagnosis as well as RA diagnosis, time from RA-ILD diagnosis (or RA diagnosis) to death, information about DMARD use during follow-up, and DAS28-CRP for the last 6 months. Second, we could not include the exposure history of individual DMARDs during follow-up as a predictor variable in the Fine-Gray regression analysis because it was a time-varying covariate. Instead, we classified each patient into good and poor RA activity control groups based on the mean DAS28-CRP value for the 6 months before the end of follow-up and found that the rate of poor control of RA activity was significantly higher in RA patients with ILD than in those without ILD. There was a negative effect of poor RA activity control on mortality in RA patients with and without ILD. However, the increased mortality observed in RA-ILD patients was not simply explained by poor RA control due to the restriction of DMARD use. Third, although lung biopsy is the gold standard for the classification of UIP patterns, we could not obtain lung biopsy samples from fibrosis lesions in RA-ILD patients. Histological confirmation of alternative diagnosis was not required in daily practice for RA-ILD patients because RA was definitively diagnosed as the cause of ILD in these cases. Additionally, there is the risk of biopsy-related complications. Considering the risk of lung biopsy and the benefit of establishing a definite diagnosis of UIP patterns, treating physicians did not perform lung biopsy in many cases. Finally, this study was conducted in our medical center located in the Kyushu area of Japan. Therefore, our findings might not be generalizable to other geographical areas.

## 5. Conclusions

Compared with the general population, the mortality rate was 2.1 times higher in patients with RA-ILD. The presence of ILD appeared to contribute to the excess death in RA patients, although all ILD cases except one were subclinical at the first diagnosis. RA-ILD patients had a mortality risk three times higher than that of RA patients without ILD. The HRCT pattern at baseline was not associated with mortality risk in patients with RA-ILD. Poor control of RA activity had a negative impact on prognosis in RA patients. Even if RA disease activity was well controlled, however, the presence of ILD was a strong predictive factor for mortality Although the generalizability of the current results must be established in future research, the present study has enhanced our understanding of the relationship between mortality risk and RA-ILD. We had expected that the early detection of ILD at RA diagnosis and during follow-up with HRCT examinations would improve patient outcomes by the timely initiation of treatment for ILD and careful therapeutic decisions for RA. However, subclinical ILD at baseline contributed to the increased risk for death in RA-ILD patients during long-term follow-up. In addition to HRCT screening for the early detection of ILD, regularly monitoring RA-ILD patients for the prompt identification of ILD progression and the development of comorbidities, such as lung cancer and acute exacerbation, is critical in improving patient outcome. A multidisciplinary approach is required for optimizing the management of each RA-ILD patient.

## Figures and Tables

**Table 1 jcm-14-01380-t001:** Baseline characteristics of RA patients with and without ILD.

	Overall	With RA-ILD	Without RA-ILD	*p **
	(*n* = 781)	(*n* = 78)	(*n* = 703)	
Age, years, mean (SD)	63.2 (12.8)	70.6 (8.1)	62.4 (13.0)	<0.001
≥18 and <45 years, number (%)	76 (9.7)	0	76 (10.8)	0.002
≥45 and <65 years, number (%)	285 (36.5)	17 (21.8)	268 (38.1)	0.004
≥65 years, number (%)	420 (53.8)	61 (78.2)	359 (51.1)	<0.001
Male, number (%)	202 (25.9)	43 (55.1)	159 (22.6)	<0.001
Duration of joint symptoms ^†^, years, mean (SD)	3.7 (7.6)	3.6 (7.8)	3.7 (7.6)	0.99
DAS28-CRP, index, mean (SD)	4.3 (1.3)	4.2 (1.1)	4.3 (1.3)	0.25
High or moderate, number (%)	696 (89.1)	67 (85.9)	629 (89.5)	0.34
Steinbrocker stages III/IV, number (%)	265 (33.9)	31 (39.7)	234 (33.3)	0.26
Anti-CCP-positive, number (%)	684 (87.6)	75 (96.2)	609 (86.6)	0.011
RF-positive, number (%)	683 (87.5)	76 (97.4)	607 (86.3)	0.003
Smoking ≥ 30 pack-years **^‡^**, number (%)	102 (13.1)	32 (41.0)	70 (10.0)	<0.001
BMI < 17, number (%)	80 (10.2)	11 (14.1)	69 (9.8)	0.24
Type 2 diabetes, number (%)	95 (12.2)	9 (11.5)	86 (12.2)	1.00
CKD stages ≥ 3, number (%)	27 (3.5)	4 (5.1)	23 (3.3)	0.34
Malignancy history, number (%)	42 (5.4)	7 (9.0)	35 (5.0)	0.18
CVD history, number (%)	72 (9.2)	14 (17.9)	58 (8.3)	0.011
Entry prior to 2011, number (%)	370 (47.4)	31 (39.7)	339 (48.2)	0.19

Data were obtained when patients were diagnosed with RA-ILD. For RA patients without ILD, data were obtained at the time of RA diagnosis. * Compared between RA patients with and without RA-ILD using Fisher’s exact probability test for categorical variables and independent-measures *t*-test for continuous variables. ^†^ Defined as duration from onset of joint symptoms and signs, which referred to patient self-reports. ^‡^ Current smokers stopped smoking according to smoking cessation instructions following RA diagnosis. RA, rheumatoid arthritis; ILD, interstitial lung disease; RA-ILD, RA-associated ILD; DAS28-CRP, 28-joint disease activity score using C-reactive protein; anti-CCP, anti-cyclic citrullinated peptide antibodies; RF, rheumatoid factor; BMI, body mass index; CKD, chronic kidney disease; CVD cardiovascular disease; SD, standard deviation.

**Table 2 jcm-14-01380-t002:** Clinical, radiological, and functional features of RA-ILD at baseline.

	RA-ILD Patients
	(*n* = 78)
Age at RA diagnosis, years, mean (SD)	68.7 (9.8)
ILD onset, number (%)	
Before RA diagnosis	3 (3.8)
Simultaneous diagnosis	65 (83.3)
Within 10 years after RA diagnosis	6 (7.7)
More than 10 years after RA diagnosis	4 (5.1)
HRCT patterns, number (%)	
Definite UIP pattern	29 (37.2)
Probable UIP pattern	14 (17.9)
Indeterminate for UIP pattern	26 (33.3)
Early UIP	5 (6.4)
NSIP/UIP	21 (26.9)
NSIP	9 (11.5)
HRCT abnormalities, number (%)	
Honeycombing	39 (50)
Traction bronchiectasis	62 (79.5)
Emphysema	36 (46.2)
Pulmonary function test ^†^, mean (SD)	
FVC, % predicted (*n* = 70)	95.3 (19.0)
FEV_1_, % predicted (*n* = 70)	113.6 (12.6)
FEV_1_/FVC ratio (*n* = 70)	125.5 (35.8)
MMF, % predicted (*n* = 70)	70.4 (30.0)
DL_CO_, % predicted (*n* = 64)	83.2 (20.3)
SpO_2_ < 90% during a 6 min walk, number (%)	1 (1.3)
Disease severity staging of ILD ^‡^, number (%)	
I	77 (98.7)
II	0
III	1 (1.3)
IV	0

Data were obtained when patients were diagnosed with RA-ILD. ^†^ Expressed as a ratio of the measured to the predicted values (% predicted), except the FEV_1_/FVC ratio. ^‡^ Determined based on PaO_2_ at rest and SpO_2_ during the 6 min walk test according to the Japanese Respiratory Society Guidelines. RA, rheumatoid arthritis; ILD, interstitial lung disease; RA-ILD, RA-associated ILD; HRCT, high-resolution computed tomography; UIP, usual interstitial pneumonia; NSIP, non-specific interstitial pneumonia; SpO_2_, oxygen saturation as measured using pulse oximeter; PaO_2_, arterial partial pressure of oxygen; FVC, forced vital capacity; FEV_1_, forced expiratory volume in one second; MMF, maximal mid-expiratory flow; DL_CO_, diffusing capacity of carbon monoxide.

**Table 3 jcm-14-01380-t003:** Mortality in RA patients with and without ILD.

	Total	With RA-ILD	Without RA-ILD
	(*n* = 781)	(*n* = 78)	(*n* = 703)
Follow-up ^†^, years, mean (SD)	10.0 (5.2)	7.4 (4.5)	10.2 (5.2)
Follow-up ^†^, years, median (IQR)	9.7 (5.9–13.9)	6.8 (3.8–10.7)	10.0 (6.1–14.1)
Lost to follow-up, number (%)	128 (16.4)	3 (3.8)	125 (17.8)
Death, number (%)	152 (19.5)	41 (52.6)	111 (15.8)
Crude incidence rate of death per 100 PYs (95% CI)	2.0 (1.7–2.0)	7.1 (5.2–10.0)	1.5 (1.0–1.9)
Cumulative incidence of death ^‡^ (%)			
1-year mortality rate (95% CI) ^§^	0.8 (0.3–1.6)	2.6 (0.5–8.1)	0.7 (0.3–1.6)
5-year mortality rate (95% CI) ^§^	4.7 (3.3–6.4)	18.6 (10.5–28.6)	3.4 (2.2–5.0)
10-year mortality rate (95% CI) ^§^	14.9 (12.3–17.8)	52.5 (38.9–64.4)	11.4 (9.0–14.1)

^†^ Follow-up began with the diagnosis of RA-ILD. For RA patients without ILD, follow-up began with the diagnosis of RA. ^‡^ Cumulative incidence (probability) of death during the follow-up period was estimated by the CIF. Gray’s test was used to compare CIF mortality estimates over time between patients with and without RA-ILD (*p* < 0.001). ^§^ Cumulative incidence of death at 1, 5, and 10 years was expressed as the estimated 1-year, 5-year, and 10-year mortality rates (%). RA, rheumatoid arthritis; ILD, interstitial lung disease; RA-ILD, RA-associated ILD; PYs, person-years; CIF, cumulative incidence function; SD, standard deviation; IQR, interquartile range; 95% CI, 95% confidence interval.

**Table 4 jcm-14-01380-t004:** Cause of death in RA patients with and without ILD.

Cause of Death	With RA-ILD	Without RA-ILD	*p* *
	(*n* = 41)	(*n* = 111)	
Age at death, years (SD)	81.2 (8.1)	77.6 (9.9)	0.042
Malignancy, number (%)	13 (31.7)	34 (30.6)	1.00
Lung cancer, number (%)	10 (24.4)	3 (2.7)	<0.001
Respiratory failure, number (%)	12 (29.3)	6 (5.4)	<0.001
Serious infection, number (%)	6 (14.6)	15 (13.5)	1.00
Cardiovascular events, number (%)	2 (4.9)	25 (22.5)	0.015
Death of old age, number (%)	5 (12.2)	13 (11.7)	0.59
Aspiration pneumonia, number (%)	2 (4.9)	5 (4.5)	1.00
Others, number (%)	1 (2.4)	13 (11.7)	0.11

* Comparison between RA patients with and without ILD using Fisher’s exact probability test for categorical variables and independent-measures t-test for continuous variables. RA, rheumatoid arthritis; ILD, interstitial lung disease; RA-ILD, RA-associated ILD; SD, standard deviation.

**Table 5 jcm-14-01380-t005:** Mortality in RA patients with and without ILD, grouped by the status of RA activity control.

	Poor Control ^†^ (*n* = 77)			Good Control ^‡^ (*n* = 704)		
	Total	With RA-ILD	Without RA-ILD	Total	With RA-ILD	Without RA-ILD
	(*n* = 77)	(*n* = 15)	(*n* = 62)	(*n* = 704)	(*n* = 63)	(*n* = 641)
Follow-up ^§^, years, mean (SD)	8.9 (5.5)	7.3 (4.8)	9.3 (5.6)	10.1 (5.2)	7.5 (4.5)	10.3 (5.2)
Follow-up ^§^, years, median (IQR)	8.1 (5.2–12.4)	6.7 (2.9–10.4)	8.2 (5.7–12.5)	9.8 (6.1–14.0)	6.9 (4.1–11.0)	10.1 (6.2–14.3)
Lost to follow-up, number (%)	9 (11.7)	0	9 (14.5)	119 (16.9)	3 (4.8)	116 (18.1)
Death, number (%)	57 (74.0)	13 (86.7)	44 (71.0)	95 (44.4)	28 (44.4)	67 (10.5)
Crude incidence rate of death per 100 PYs (95% CI)	8.3 (6.4–11.0)	11.9 (6.9–20.5)	7.7 (5.7–10.3)	1.3 (1.1–1.6)	5.9 (4.1–9.0)	1.0 (0.8–1.3)

^†^ Defined as high or moderate disease activity using DAS28-CRP values in the 6 months before the end of follow-up. ^‡^ Defined as remission or low disease activity using DAS28-CRP values in the 6 months before the end of follow-up. ^§^ Follow-up began with the diagnosis of RA-ILD. For RA patients without ILD, follow-up began with the diagnosis of RA. RA, rheumatoid arthritis; ILD, interstitial lung disease; RA-ILD, RA-associated ILD; DAS28-CRP, 28-joint disease activity score using C-reactive protein; PYs, person-years; SD, standard deviation; IQR, interquartile range; 95% CI, 95% confidence interval.

**Table 6 jcm-14-01380-t006:** Predictive factors for death in overall RA patients.

Variables	Unadjusted HRs	*p **	Adjusted HRs	*p **
	(95% CI)		(95% CI)	
Age per additional year	1.09 (1.06–1.11)	<0.001	1.08 (1.05–1.10)	<0.001
Male vs. female	1.95 (1.41–2.71)	<0.001	1.61 (1.02–2.53)	0.042
Disease duration per additional year	1.02 (1.00–1.03)	0.026	0.99 (0.97–1.01)	0.46
DAS28-CRP per additional unit	0.98 (0.87–1.11)	0.75	–	–
Stages III/IV vs. I/II	1.31 (0.96–1.79)	0.090	0.98 (0.69–1.39)	0.91
Anti-CCP-positive vs. Anti-CCP-negative	3.00 (1.41–6.40)	0.005	2.87 (1.25–6.56)	0.013
RF-positive vs. RF-negative	2.19 (1.12–4.30)	0.022	1.07 (0.50–2.28)	0.87
RA-ILD, yes vs. no	5.84 (4.08–8.34)	<0.001	2.97 (1.95–4.53)	<0.001
Smoking > 30 PYs vs. no history	2.40 (1.64–3.51)	<0.001	1.31 (0.78–2.17)	0.31
BMI < 17, yes vs. no	2.89 (2.00–4.17)	<0.001	3.07 (2.10–4.49)	<0.001
Type 2 diabetes, yes vs. no	0.51 (0.27–0.97)	0.040	0.52 (0.27–1.01)	0.054
CKD stage ≥ 3, yes vs. no	1.98 (1.10–3.56)	0.020	1.52 (0.72–3.19)	0.27
Malignancy history, yes vs. no	1.34 (0.69–2.60)	0.38	–	–
CVD history, yes vs. no	1.57 (0.95–2.57)	0.080	1.14 (0.65–1.99)	0.65
Entry before 2011, yes vs. no	1.48 (1.04–2.10)	0.030	1.42 (0.98–2.07)	0.65

* Univariable and multivariable Fine-Gray competing risk regression analyses were conducted for overall RA patients. RA, rheumatoid arthritis; ILD, interstitial lung disease; RA-ILD-RA-associated ILD; anti-CCP, anti-cyclic citrullinated peptide antibodies; RF, rheumatoid factor; BMI, body mass index; CKD, chronic kidney disease; CVD, cardiovascular disease; PY, pack-years; HR, hazard ratio; 95% CI, 95% confidence interval.

**Table 7 jcm-14-01380-t007:** Predictive factors for mortality in RA-ILD patients.

Variables	Unadjusted HR	*p **	Adjusted HR	*p **
	(95% CI)		(95% CI)	
Age per additional year	1.07 (1.03–1.12)	0.001	1.08 (1.03–1.13)	0.002
Male vs. female	1.47 (0.79–2.71)	0.22	–	–
Disease duration per additional year	0.99 (0.95–1.02)	0.43	–	–
DAS28-CRP per additional unit	1.04 (0.81–1.33)	0.77	–	–
Steinbrocker stages III/IV vs. I/II	0.80 (0.43–1.47)	0.47	–	–
Anti-CCP-positive vs. Anti-CCP-negative	0.70 (0.15–3.46)	0.87	–	–
RF-positive vs. RF-negative	0.27 (0.07–1.07)	0.060	0.40 (0.09–1.78)	0.23
HRCT patterns of ILD				
NSIP	1 (reference)	–	–	–
Early UIP	0.51 (0.14–1.87)	0.31	–	–
NSIP/UIP	0.51 (0.17–1.51)	0.22		
Probable UIP	0.42 (0.13–1.36)	0.15	–	–
Definite UIP	0.66 (0.29–1.53)	0.33	–	–
HRCT abnormalities				
Honeycomb, yes vs. no	1.43 (0.77–2.67)	0.26	–	–
Traction bronchiectasis, yes vs. no	0.80 (0.42–1.52)	0.49	–	–
Emphysema, yes vs. no	1.06 (0.58–1.94)	0.85	–	–
Smoking > 30 PYs vs. no history	1.54 (0.83–2.85)	0.17	–	–
BMI < 17.0, yes vs. no	4.59 (1.64–12.83)	0.004	5.29 (2.28–12.27)	<0.001
Type 2 diabetes, yes vs. no	0.17 (0.02–1.46)	0.11	–	–
CKD stage ≥ 3, yes vs. no	0.48 (0.05–4.43)	0.51	–	–
Malignancy history, yes vs. no	1.58 (0.84–2.96)	0.16	–	–
CVD history, yes vs. no	0.21 (0.07–0.68)	0.009	0.26 (0.08–0.85)	0.030
Entry before 2011, yes vs. no	1.34 (0.71–2.51)	0.36	–	–

* Univariable and multivariable Fine-Gray competing risk regression analyses were conducted for RA-ILD patients. RA, rheumatoid arthritis; ILD, interstitial lung disease; RA-ILD, RA-associated ILD; anti-CCP, anti-cyclic citrullinated peptide antibodies; RF, rheumatoid factor; HRCT, high-resolution computed tomography; UIP, unusual interstitial pneumonia; NSIP, non-specific interstitial pneumonia; BMI, body mass index; CKD, chronic kidney disease; CVD, cardiovascular disease; PY, pack-years; HR, hazard ratio; 95% CI, 95% confidence interval.

## Data Availability

All data supporting the findings are available from the Human Research Ethics Committee of National Hospital Organization Kumamoto Saishun Medical Center for all interested researchers who meet the criteria for access to confidential data. Because these data include patients’ personal information, the Committee does not recommend that such data be made unnecessarily public. Please contact Mr. Masahiro Hamaguchi, the Control Manager of the Committee, at 616-syol@mail.hosp.go.jp to request the data.

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
