# Peer review of "Mortality and Predictive Factors for Death Following the Diagnosis of Interstitial Lung Disease in Patients with Rheumatoid Arthritis: A Retrospective, Long-Term Follow-Up Study"

_jcm, 2025, doi:10.3390/jcm14041380_

Round 1
Reviewer 1 Report
Comments and Suggestions for Authors
In this retrospective study, the authors examined mortality rates and factors predictive of death in patients with rheumatoid arthritis (RA) diagnosed with interstitial lung disease (ILD). They found that patients with RA-ILD had significantly higher mortality rates than patients with RA without ILD and identified some predictive factors of mortality.
Comments:
1. Patients were analyzed over a large period of time. As the authors point out, the diagnosis of rheumatoid arthritis was made using the 1987 and 2010 criteria. These criteria are known to differ widely in terms of specificity and sensitivity. This may have introduced a selection bias that the authors should comment on in more detail.
2. The diagnosis of the type of ILD was based on images obtained by HR-CT. As anticipated by the authors, radiological pictures could sometimes be similar between UIP and NSIP. Histopathological analysis would have given more significance to the data presented. Please comment on this point.
3. Smoking habit was significantly more represented in patients without ILD. This suggests that smoking is a protective factor or, alternatively, that patients with ILD are less likely to smoke. The authors should provide an explanation for this result.
4. Patients with ILD made less use of methotrexate (MTX) than those without ILD because of the fear of a pro-fibrotic effect of MTX on the lung. Although this fear is questionable based on recent data that do not correlate MTX use with the risk of worsening ILD, this may have produced a bias in the results obtained. this point also deserves further comment.
5. The authors should introduce data on the treatment of patients with ILD with alternative therapies to MTX (biologics, JAKis) to clarify any differences between the two groups considered and how these treatments may affect mortality rates.
Author Response
Response to Reviewer 1
We are most grateful to Reviewer 1 for their valuable comments and the time and energy spent in reviewing our manuscript. We have made the requested changes and added new information to the manuscript in response to the reviewer’s insightful comments. All alterations are highlighted in red text in the revised manuscript. We are confident that the manuscript has benefited from the reviewer’s useful comments and suggestions.
Below are point-by-point replies to Reviewer 1’s comments.
Comment 1: Patients were analyzed over a large period of time. As the authors point out, the diagnosis of rheumatoid arthritis was made using the 1987 and 2010 criteria. These criteria are known to differ widely in terms of specificity and sensitivity. This may have introduced a selection bias that the authors should comment on in more detail.
Author Response: We thank the reviewer for this insightful comment. First, I must apologize for our careless error in the description of the exclusion criteria in the Materials and Methods of the previous version. For inclusion in the study, participants were required to meet the 2010 American College of Rheumatology (ACR)/European League Against Rheumatism (EULAR) criteria for diagnosis of RA (lines 87–89). Data on joint involvement, serology (RF and ant-CCP), acute-phase reactants (CRP and ESR), and duration of symptoms, which are required for the 2010 ACR/ EULAR diagnosis of RA, were all available in our database. For patients who were diagnosed with RA or RA-ILD before a commercially available ELISA kit was released, we used sera collected at the time of diagnosis and stored at –80°C. Detailed methods for the measurements of anti-CCP are available in our previous publication (ref. 40 and lines 105–108). We have added a participant flow diagram including the number of patients who were included or excluded from this study as well as the number of events for the primary outcome (Supplementary Figure S1 and lines 212 and 540–543).
Comment 2: The diagnosis of the type of ILD was based on images obtained by HR-CT. As anticipated by the authors, radiological pictures could sometimes be similar between UIP and NSIP. Histopathological analysis would have given more significance to the data presented. Please comment on this point.
Author Response: We appreciate this important comment. We understand that lung biopsy is the gold standard for the classification of UIP patterns. However, lung biopsy samples from fibrosis lesions were available for only a few patients with RA-ILD in our cohort. In many cases, treating physicians did not perform a biopsy because of concerns about the risk of biopsy-related complications. Additionally, histological confirmation was not required for RA-ILD patients in daily practice because RA was definitively diagnosed as the cause of ILD. For this study, we collected HRCT images taken at the time of RA or RA-ILD diagnosis from all participants, which were viewed in random order and independently by three experts with broad experience in reading abnormalities on pulmonary HRCT, especially of parenchymal and airway lung diseases. We are confident with the HRCT-based diagnoses in this study. In response to the reviewer’s comment, we included the lack of histopathological data in the Discussion section as one of the limitations of this study in the revised manuscript (lines 508–514).
Comment 3: Smoking habit was significantly more represented in patients without ILD. This suggests that smoking is a protective factor or, alternatively, that patients with ILD are less likely to smoke. The authors should provide an explanation for this result.
Author Response: We acknowledge the reviewer’s comment on this point. As shown in Table 1, smoking history was significantly associated with the presence of RA-ILD (41.0% in RA-ILD patients vs. 10.0% in RA patients without ILD, p < 0.001) (Table 1 and lines 224–226). Smoking, male sex, advanced age, and RA-related autoantibodies are recognized as patient-level risk factors for developing RA-ILD. We have added this information in the Introduction section of the revised manuscript (lines 55 and 56), and a new reference has been included (ref. 18). These factors were observed in RA-ILD patients at a significantly higher rate compared with RA patients without ILD in the present study (Table 1).
Comment 4: Patients with ILD made less use of MTX than those without ILD because of the fear of a pro-fibrotic effect of MTX on the lung. Although this fear is questionable based on recent data that do not correlate MTX use with the risk of worsening ILD, this may have produced a bias in the results obtained. this point also deserves further comment.
Author Response: We thank the reviewer for this insightful comment. As the reviewer pointed out, the restriction of DMARD use, especially MTX use, in RA-ILD patients, may have induced a higher rate of poor control of RA activity in this patient group compared with RA patients without ILD (lines 329–332). Concern about lung injury is greatest in patients receiving MTX, but a causal effect between MTX and fibrotic ILD has not been established. The overall risk of worsening lung disease attributable to MTX is uncertain. Recent studies do not support the idea that RA-ILD progression is increased in patients receiving MTX. The 2021 ACR guidelines for treatment of RA include a conditional recommendation for MTX for the treatment of RA patients with clinically diagnosed mild and stable airway or parenchymal lung disease who have moderate to high RA disease activity. The appropriate use of MTX considering RA disease activity would be important for the management of RA-associated ILD. The use of IL-6 inhibitors and abatacept may be considered in the treatment for RA-ILD patients because concomitant use of MTX is not necessarily required for these DMARD therapies. In response to the reviewer’s comment, we have included the above-mentioned information in the Discussion section of the revised manuscript (lines 461–473). Several references have been added (refs. 30 and 70–73). In the previous version of the manuscript, we had described differences in DMARD use between RA-ILD patients and RA patients without ILD as a limitation of this study as follows: we could not include exposure history of individual DMARDs during follow-up as a predictor variable in the Fine-Gray regression analysis because it was a time-varying covariate. Instead, we classified each patient into good and poor RA activity control groups based on the mean DAS28-CRP value for the 6 months before the end of follow-up and found that the rate of poor control of RA activity was significantly higher in RA patients with ILD than in those without ILD. There was a negative effect of poor RA activity control on mortality in overall RA patients. However, the increased mortality in RA-ILD patients was not explained simply by poor RA control due to the restriction of MTX use. We modified several sentences in this part in the revised version (lines 499–508). Data regarding comparisons of mortality between RA patients with and without ILD grouped by RA disease activity control status between groups are shown in the Results section (lines 327–340) and Table 5.
Comment 5: The authors should introduce data on the treatment of patients with ILD with alternative therapies to MTX (biologics, JAKis) to clarify any differences between the two groups considered and how these treatments may affect mortality rates.
Author Response: We appreciate this comment. As shown in Supplementary Table S1 of the previous version of the manuscript, RA-ILD patients were more often prescribed other csDMARDs instead of MTX. Additionally, IL-6 inhibitors and abatacept were prescribed more frequently for RA-ILD patients than for RA patients without ILD. The rates of TNF inhibitor and JAK inhibitor use were similar between the two patient groups (Supplementary Table S1 and lines 314–323). In response to the reviewer’s comment, we have added new information regarding the use of DMARDs in RA-ILD patients grouped by MTX use in the revised version (Supplementary Table S2 and line 539). MTX users more often received other csDMARDs, and rates of TNF inhibitor use were significantly higher in MTX users than in non-MTX users, because TNF inhibitors are recommended to be used concomitantly with MTX. There were no significant differences in the rates of use of other DMARD (lines 323–326).

Reviewer 2 Report
Comments and Suggestions for Authors
This manuscript summarizes the mortality and predictive factors for death in patients with rheumatoid arthritis (RA) with and without interstitial lung disease (ILD). It provides valuable insights into the impact of RA-ILD on patient survival and highlights the role of high-resolution computed tomography (HRCT) patterns and RA disease activity control. However, several issues need to be addressed to improve clarity and strengthen the comprehensiveness of this study.
Comments:
1. More details are needed on how HRCT patterns were classified and assessed. Were inter-reader reliability or kappa statistics used to validate image interpretation?
2. The regression analysis results should be more explicitly linked to clinical implications. How do these findings influence treatment strategies for RA-ILD?
3. Consider providing a clearer explanation of the statistical significance of HRCT patterns in RA-ILD mortality.
4. Expand on the potential clinical applications of HRCT screening for ILD.
5. The hypothesis that poor RA disease control contributes to higher mortality should be supported with more discussion on treatment modifications and possible therapeutic strategies.
Author Response
Response to Reviewer 2
We are most grateful to Reviewer 2 for their valuable comments and the time and energy spent in reviewing our manuscript. We have made the requested changes and added new information to the manuscript in response to the reviewer’s insightful comments. All alterations are highlighted in red text in the revised manuscript. We are confident that the manuscript has benefited from the reviewer’s useful comments and suggestions.
Below are point-by-point replies to Reviewer 2’s comments.
Comment 1: More details are needed on how HRCT patterns were classified and assessed. Were inter-reader reliability or kappa statistics used to validate image interpretation?
Author Response: We greatly appreciate this insightful comment. HRCT images were viewed in random order and independently following three observers (one board-certified radiologist and two board-certified pulmonologists) who were blinded to the patients’ clinical status and PFT results. Final decisions were made following discussion if there was disagreement. We did not determine the inter-reader agreement by kappa statistics, because all observers were very experienced readers of abnormalities on pulmonary HRCT, especially of parenchymal and airway lung diseases. HRCT abnormalities included the following findings: bronchiectasis or bronchiolectasis, bronchial wall thickening, centrilobular micro nodules and branching structure, cysts, ground-glass opacity, intralobular reticular opacity, airspace consolidation, honeycombing, traction bronchiectasis, architectural distortion, and emphysema. The distribution of HRCT abnormalities was evaluated according to six zones (upper, middle, and lower zones; involvement in at least 5% of any lung zone). We also examined whether these abnormalities had a predominantly subpleural or non-subpleural distribution. We have added these details in the Materials and Methods section of the revised version (lines 134–145). As mentioned in the previous version, HRCT patterns were classified according to the updated official American Thoracic Society/European Respiratory Society/Japanese Respiratory Society/Latin American Thoracic Society clinical practice guidelines for idiopathic pulmonary fibrosis. We further classified the “indeterminate for UIP pattern” into early UIP and NSIP/UIP based on the modified HRCT classification for RA-ILD reported by Yamakawa et al. (lines 146–167).
Comment 2: The regression analysis results should be more explicitly linked to clinical implications. How do these findings influence treatment strategies for RA-ILD?
Author Response: We thank the reviewer for this comment regarding the clinical implications of our results. We agree with the reviewer’s suggestion to discuss how our findings can influence treatment strategies for RA-ILD. Using multivariable Fine-Gray regression analysis, we identified the presence of ILD, advanced age, and low BMI as strong predictive factors for mortality in RA patients. Old age and low BMI are generally recognized as predictive factors for mortality; our concern is the presence of ILD. In the present study, all ILD cases except one were subclinical at the first diagnosis. Nevertheless, the mortality rate was 2.1 times higher in RA-ILD patients compared with the general population during long-term follow-up with a mean of 10 years, whereas excess death was not observed in RA patients without ILD (lines 282–284). Among our RA-ILD patients, the most common cause of death was respiratory failure due to an acute exacerbation of ILD and lung cancer. Death due to lung cancer and respiratory failure occurred more frequently in RA-ILD patients than in RA patients without ILD (Table 4). Some RA patients with ILD are known to experience acute exacerbation with sudden progression of fibrosis and short-term mortality. Unfortunately, it is difficult to predict which patients with RA-ILD are at high risk for ILD progression. Regular monitoring of RA-ILD patients is important for prompt identification of ILD progression and early intervention for preserving lung function. Recently, we reported that the standardized incidence ratio of lung cancer in our RA cohort compared with that in the general population was 2.53 (95% CI, 1.29–3.77) for male patients. The presence of ILD was a strong predictive factor for lung cancer occurring in RA patients over time. During follow-up, increased lung cancer-related mortality was observed compared with non-RA patients (crude incidence rate: 0.29 per PY vs. 0.10 per PY). The presence of combined pulmonary fibrosis and emphysema was identified as an independent predictive factor for increased lung cancer-related mortality. Non-surgical cancer treatment includes chemotherapy and radiation therapy, which are well known to exacerbate pulmonary fibrosis. Limited options for lung cancer treatment and treatment-associated complications may be associated with the increased lung cancer-related mortality in RA-ILD patients. To clarify these points, we have added one paragraph in the Discussion section of the revised version (lines 474–493). New references were included (refs. 68 and 74–76). Regular monitoring for quick identification of ILD progression as well as development of comorbidities such as lung cancer and acute exacerbation are critical to improve patient prognosis. A multidisciplinary approach is required for optimizing the management of each RA-ILD patient. Accordingly, we modified the Conclusions section in the revised version of the manuscript (lines 531–535).
Comment 3: Consider providing a clearer explanation of the statistical significance of HRCT patterns in RA-ILD mortality.
Author response: We thank the reviewer for this comment. In the previous version, we introduced results from a number of previous studies (lines 419–424). The effect of HRCT pattern on mortality in RA-ILD patients remains controversial. Recently, Yamakawa et al. developed a modified HRCT classification for RA-ILD, in which the “indeterminate for UIP” pattern was classified into early UIP and NSIP/UIP patterns. Using the modified HRCT classification, Yamakawa et al. showed that their RA-ILD cohort included NSIP/UIP (27%), definite UIP (21%), probable UIP (20%), NSIP (14%), and early UIP (3%). In that study, HRCT pattern was not a predictive factor for mortality in multivariable Cox regression analysis. In the present study, we followed the modified HRCT classification by Yamakawa et al. and found that, in our cohort, the most common HRCT pattern was definite UIP (37.2%) followed by NSIP/UIP (26.9%), probable UIP (17.9%), NSIP (11.5%), and early UIP (6.4%). HRCT pattern was not related to mortality risk in our RA-ILD patients. In both studies, most patients with the “indeterminate for UIP” pattern were classified as having the NSIP/UIP pattern. In other previous studies, the NSIP/UIP pattern might have been diagnosed simply as the NSIP pattern or the UIP pattern, which could have resulted in different conclusions regarding the effect of HRCT pattern on mortality risk in RA-ILD patients. In our statistical analyses, the presence of ILD was a strong predictive factor for death, and the NSIP pattern did not have a significantly better prognosis than the UIP, NSIP/UIP, or early UIP pattern, which suggests that, regardless of the HRCT pattern, RA patients with ILD are at increased risk of death. To clarify these points, we have modified this paragraph (lines 424–441).
Comment 4: Expand on the potential clinical applications of HRCT screening for ILD.
Author Response: We thank the reviewer for this suggestion. We had expected that early detection of ILD at RA diagnosis and during follow-up by HRCT examinations would improve patient outcomes by timely initiation of treatment of ILD and careful therapeutic decisions for RA. However, subclinical ILD at baseline contributed to the increased risk for death in RA-ILD patients during long-term follow-up. HRCT pattern at baseline was not associated with mortality risk in patients with RA-ILD. Considering these results as well as the main cause of death, RA-ILD patients need regular monitoring for quick identification of ILD progression as well as development of comorbidities such as lung cancer and acute exacerbation. We have included these comments in the Conclusions section of the revised manuscript (lines 527–535). As we mentioned in the Discussion section, several studies have reported that the extent of lung involvement, rather than the baseline UIP pattern, independently predicts mortality (lines 421–423). Jacob et al. showed that individual extents of fibrosis, reticulation, and honeycombing on HRCT scans can predict the develop of progressive fibrosis and poor outcomes in RA-ILD patients. In this respect, HRCT screening for ILD at RA diagnosis and during follow-up can provide insight into the extent of these abnormalities. We have included this information in the Discussion section (lines 441–445 and ref. 67).
Comment 5: The hypothesis that poor RA disease control contributes to higher mortality should be supported with more discussion on treatment modifications and possible therapeutic strategies.
Author Response: Thank you very much for this insightful comment.We entirely agree with the reviewer’s comment for the need for more discussion on treatment modifications and possible therapeutic strategies. In response to this comment, we have added the following discussion to the paragraph in the revised manuscript: in the present study, poor control of RA activity was associated with the increased risk of death. Restriction of DMARD use in RA-ILD patients may have resulted in the higher rate of poor control of RA activity in this patient group compared with RA patients without ILD. Concern about lung injury have been greatest in patients receiving MTX, but a causal effect between MTX and fibrotic ILD has not been established. The overall risk of worsening lung disease attributable to MTX is uncertain. Recent studies do not support the idea that RA-ILD progression is increased in patients receiving MTX. The 2021 ACR guideline for treatment of RA includes a conditional recommendation for MTX for the treatment of RA patients with clinically diagnosed mild and stable airway or parenchymal lung disease who have moderate to high disease activity. The appropriate use of MTX considering RA disease activity would be important for the management of RA-associated ILD. Additionally, the use of IL-6 inhibitors and abatacept may be considered in the treatment of RA-ILD patients because concomitant use with MTX is not necessarily required for these DMARD therapies (lines 460–473). In addition, new references have been included (refs. 30 and 70–73).

Reviewer 3 Report
Comments and Suggestions for Authors
This manuscript explores predictive factors for mortality in interstitial lung disease (ILD) among rheumatoid arthritis (RA) patients based on a long-term follow-up study with a large sample size. It is of outstanding quality, informative, insightful, well-written, and well-organised. The findings, analyses, and outcomes are appropriate, and the identified predictive factors align with the major factors reported in the literature.
Comments.
1. Flow diagram: Consider creating a flow diagram following the CONSORT guidelines to summarise the study procedure, from patient recruitment to endpoint assessment. This would enhance clarity and facilitate understanding of the study design.
2. Lines 39–42: Suggest specifying which tissues and organs are most commonly affected by RA, to provide a clearer picture of its systemic impact. For example, mention synovial joints, lungs, skin, heart, and blood vessels.
3. Lines 54–55: Suggest clarifying why ILD remains a major contributor to mortality in RA. Possible factors include late diagnosis, lack of targeted therapies, rapid disease progression, or complications such as pulmonary hypertension and respiratory failure.
4. Lines 75–76: Suggest specifying how RA disease activity was assessed in the study. For example, did the study use clinical scores (e.g., DAS28, CDAI), inflammatory biomarkers (e.g., CRP, ESR), or imaging assessments? Adding this information would improve clarity.
5. Table 1: Suggeest including age subgroups (e.g., young adults, middle-aged, elderly), as mean age differences between groups were statistically significant. This additional stratification would make the table more informative.
6. Lines 372–374: Suggest clarifying whether subclinical ILD at diagnosis influenced treatment decisions or patient outcomes. Did early detection lead to earlier intervention or different therapeutic strategies?
Author Response
Response to Reviewer 3
We are most grateful to Reviewer 3 for their valuable comments and the time and energy spent in reviewing our manuscript. We have made the requested changes and added new information to the manuscript in response to the reviewer’s insightful comments. All alterations are highlighted in red text in the revised manuscript. We are confident that the manuscript has benefited from the reviewer’s useful comments and suggestions.
Below are point-by-point replies to Reviewer 3’s comments.
Comment 1: Flow diagram: Consider creating a flow diagram following the CONSORT guidelines to summarize the study procedure, from patient recruitment to endpoint assessment. This would enhance clarity and facilitate understanding of the study design.
Author Response: We thank the reviewer for this insightful comment. I understand that the CONSORT flow diagram is useful to report available information about the total number of participants at each stage of the randomized clinical trial, with reasons for non-enrollment or loss to follow-up. Since our study was a retrospective follow-up study, we followed the STROBE statement, which was developed to improve the quality of reporting observational studies. Although the STROBE guidelines also recommend the use of a flow diagram, they do not propose a specific format for the diagram. Since we agree with the reviewer’s comment on the use of the flow diagram, we have added a participant flow diagram in the revised manuscript (Supplementary Figure S1 and lines 212 and 540–543).
Comment 2: Lines 39–42: Suggest specifying which tissues and organs are most commonly affected by RA, to provide a clearer picture of its systemic impact. For example, mention synovial joints, lungs, skin, heart, and blood vessels.
Author Response: We thank the reviewer for this suggestion. We revised the initial part of the Introduction section as follows: Rheumatoid arthritis (RA) is a chronic immune-mediated rheumatic disease that primarily involves multiple synovial joints; however, systemic inflammation associated with RA can cause extra-articular damage to various tissues and organs such as the lungs, heart, blood vessels, renal system, skin, nervous system, and eyes (lines 38–41).
Comment 3: Lines 54–55: Suggest clarifying why ILD remains a major contributor to mortality in RA. Possible factors include late diagnosis, lack of targeted therapies, rapid disease progression, or complications such as pulmonary hypertension and respiratory failure.
Author Response: We greatly appreciate this suggestion. We agree with the reviewer’s opinion regarding the reasons for poor outcome in RA patients with ILD. Accordingly, we have added these risk factors in the Introduction section of the revised manuscript as follows: Delayed diagnosis, paucity of clinical trials of biological or targeted drugs, the existence of cases with rapidly progressive fibrosis, and serious complications (such as lung cancer, pulmonary infection, pulmonary hypertension, and respiratory failure) may cause poor outcomes associated with RA-ILD (lines 61–64). A new reference has been added (ref. 30). In fact, in the present study, the most common cause of death was respiratory failure due to an acute exacerbation of ILD (29.3%) and lung cancer (24.4%), and both complications occurred more frequently in RA-ILD patients than in RA patients without ILD (Table 4).
Comment 4: Lines 75–76: Suggest specifying how RA disease activity was assessed in the study. For example, did the study use clinical scores (e.g., DAS28, CDAI), inflammatory biomarkers (e.g., CRP, ESR), or imaging assessments? Adding this information would improve clarity.
Author Response: We thank the reviewer for this suggestion. In the present study, we used DAS28-CRP values. Based on the mean of DAS28-CRP values measured for 6 months prior to the end of follow-up, each patient was classified into the group having good control of RA activity (remission or low disease activity [DAS28-CRP <2.7]) or the group with poor control of RA activity (moderate or high disease activity [DAS28-CRP ≥2.7]). In the previous version, we had included this information in the Materials and Methods section (lines 168–174) and in the footnote of Table 5. In response to the reviewer’s comment, we have added this information to the last paragraph of the Introduction section (lines 81 and 82).
Comment 5: Table 1: Suggest including age subgroups (e.g., young adults, middle-aged, elderly), as mean age differences between groups were statistically significant. This additional stratification would make the table more informative.
Author Response: We greatly appreciate this insightful comment. We have included information on the number (%) of RA patients grouped by age. The rates of patients in each age group were significantly different between RA-ILD patients and RA patients without ILD (Table 1).
Comment 6: Lines 372–374: Suggest clarifying whether subclinical ILD at diagnosis influenced treatment decisions or patient outcomes. Did early detection lead to earlier intervention or different therapeutic strategies?
Author Response: We thank the reviewer for this suggestion. In the present study, all ILD cases except one were subclinical at the first diagnosis, and therefore we could not compare DMARD use or outcomes between subclinical ILD and clinically apparent ILD. As the reviewer pointed out, we expected that early detection of ILD at RA diagnosis and during follow-up would improve patient outcomes by timely initiation of ILD treatment and careful therapeutic decisions for RA. However, the crude incidence rate of death was significantly higher in RA-ILD patients than in RA patients without ILD (Table 3). The standardized mortality rates in our RA cohort compared with the general population in Japan were 1.32 (95% CI, 1.11–1.53) for overall RA patients, 2.09 (95% CI, 1.45–2.73) for RA-ILD patients, and 1.16 (95% CI, 0.95–1.38) for RA patients without ILD (lines 282–284). Approximately 90% of RA patients without ILD received MTX as monotherapy or in combination with other DMARDs, whereas more than 40% of RA-ILD patients never received MTX, most likely because of the concern for ILD exacerbation. IL-6 inhibitors and abatacept were prescribed more frequently for RA-ILD patients than for RA patients without ILD. The rates of TNF inhibitor and JAK inhibitor use were similar between the two patient groups (Table S1). The rate of poor control of RA activity was significantly higher in RA patients with ILD than in those without ILD (19.2% vs. 8.8%, p = 0.008), which may be explained by the restriction of DMARD use, especially MTX use, in RA-ILD patients. As shown in Table 5, the crude incidence rate of death events was significantly higher in the poor RA control group compared with the good RA control group. However, RA-ILD patients had a higher incidence rate of death than RA patients without ILD in the good RA control group. Thus, the increased mortality in RA-ILD patients was not explained simply by poor RA control due to the restriction of DMARD use. These findings indicated that the presence of ILD is associated with the increased mortality in RA-ILD patients, even in those patients whose pulmonary abnormalities are detected early. In our RA-ILD cohort, the most common cause of death was respiratory failure due to an acute exacerbation of ILD (29.3%) and lung cancer (24.4%) (Table 4). In addition to HRCT screening for early detection of ILD, regular monitoring of RA-ILD patients for prompt identification of ILD progression and development of comorbidities such as lung cancer and acute exacerbation is critical to improve patient outcomes. A multidisciplinary approach is required for optimizing the management of each RA-ILD patient. In response to the reviewer’s comment, we revised the Conclusions section (lines 527–535).

Round 2
Reviewer 1 Report
Comments and Suggestions for Authors
The authors responded exhaustively to my comments and modified the manuscript accordingly.
Reviewer 2 Report
Comments and Suggestions for Authors
All my concerns have been addressed.